# Central and Effector Memory Human CD4^+^ and CD8^+^ T Cells during Cutaneous Leishmaniasis and after In Vitro Stimulation with *Leishmania (Viannia) braziliensis* Epitopes

**DOI:** 10.3390/vaccines11010158

**Published:** 2023-01-11

**Authors:** Beatriz Coutinho de Oliveira, Ailton Alvaro da Silva, Marton Kaique de Andrade Cavalcante, Maria Edileuza Felinto de Brito, Maria Carolina Accioly Brelaz de Castro, Vanessa Lucília Silveira de Medeiros, Rafael de Freitas e Silva, Valéria Rêgo Alves Pereira

**Affiliations:** 1Cleveland Clinic Lerner College of Medicine, Cleveland, OH 44195, USA; 2Therapeutic Innovation Graduate Program (PPGIT), Federal University of Pernambuco, Recife 50670-901, Brazil; 3Immunology Department, Aggeu Magalhães Institute, Recife 50740-465, Brazil; 4Academic Center of Vitória, Federal University of Pernambuco, Vitória de Santo Antão 55608-680, Brazil; 5Dermatology Clinics, Tropical Medicine Department, Clinics Hospital, Federal University of Pernambuco, Recife 50670-901, Brazil; 6Medical Faculty of Recife, Medical Sciences Center, Federal University of Pernambuco, Recife 50670-901, Brazil; 7Department of Natural Sciences, University of Pernambuco, Garanhuns Campus, Garanhuns 55294-902, Brazil

**Keywords:** Cutaneous Leishmaniasis, *Leishmania (Viannia) braziliensis*, peptides, vaccine, memory T cells

## Abstract

Cutaneous Leishmaniasis (CL) is a Neglected Tropical Disease characterized by skin ulcers caused by *Leishmania* spp. protozoans and there is no safe and effective vaccine to reduce its negative consequences. In a previous work by our group, we identified T cell epitopes of *Leishmania (Viannia) braziliensis* which stimulated patients’ T cells in vitro. In the present work, the peptides were tested as two pools for their ability to rescue memory T cells during natural infection by Leishmania. We analyzed the frequency of central memory (TCM, CD45RA-CD62L+) and effector memory (TEM, CD45RA + CD62L-) cells during active CL and post-treatment. In parallel, we investigated cell proliferation levels and the cytokines produced after stimulation. Interestingly, we observed higher frequencies (%) in CD4+ TEM during CL, and CD8+ TEM and CD8+ TCM during CL and post-treatment. Cell proliferation was increased, and a significant difference in expression was observed on T-bet and RORγT. Besides that, IFN-γ, IL-2, and IL-10 were detected in patient samples. Collectively, this dataset suggests that during CL there is an increase in the frequency of TCM and TEM, especially in the CD8 compartment. These results indicate a potentially immunogenic profile of the peptide pools, which can support the development of anti-Leishmania formulations.

## 1. Introduction

Cutaneous Leishmaniasis (CL) is part of a group of skin diseases caused by protozoans of the genus *Leishmania* spp. and is transmitted to humans through the blood meal of infected sandflies. Disease estimates suggest that CL is present in 98 countries, with 87 of them being endemic. Afghanistan, Colombia, Iran, Tunisia, and Brazil are some of the most affected countries, where approximately 2 million cases occur each year [1,2]. CL is characterized by the formation of one or more lesions, usually painless, with well-defined borders and a granulomatous background. When left untreated, they may heal spontaneously over a period of months to years or may persist for a longer time. One of the species involved with this clinical manifestation is *Leishmania (Viannia) braziliensis*, which is responsible for most cases occurring in Latin America, especially in Brazil, where it has been reported in all states [3,4,5]. 

In the absence of a vaccine or immune active formulations, the pentavalent antimonials are the main treatment of CL but are limited by high cost, high toxicity, adverse effects, and parasite resistance through decades of use [5,6,7]. Therefore, it is highly necessary to search for new approaches to control or eliminate CL. Studies and cost-benefit analyses show that the use of a vaccine would be more advantageous than the production of new drugs [8]. The lack of a vaccine is explained by the complexities of parasites and host immunity, which are not completely understood in the context of the disease [9,10]. Previously used classical vaccine approaches have led to the identification of several molecules. Some made it up to the clinical trial phases and one of them had controversial results when they had to induce a protective immune response [11]. These studies were able to identify antigens based on their ability to stimulate a potent antibody response, however, they failed in stimulating cellular immune responses, which are essential for parasite elimination [12,13]. 

T cells are fundamental for Leishmania immunity, and it is generally accepted that the difference between resistance or susceptibility to infection is associated with the level of expansion of CD4+ Th1 and CD4+ Th2 cells, respectively [14,15,16]. Th1 cells have their differentiation stimulated by the cytokines IL-12 and interferon-gamma (IFN-γ) and express the transcription factor T-bet. Th1 cells producing IFN-γ are, so far, the best correlate of protection once IFN-γ activates macrophages to eliminate parasites. Cells from the Th2 profile are stimulated by IL-4 and express the transcription factor GATA-3 [17,18]. CD8+ T cells are essential to control infection as proved by an experiment where mice susceptible to *L. major* infection were transiently depleted for CD4+ T cells and the CD8+ T cells kept the animals resistant to infection [19,20]. Part of this protective role of CD8+ T cells is linked to their ability to switch the Th2 response to the Th1 response [21]. However, the exacerbation of inflammation in experimental and human leishmaniasis was previously reported with an augmented release of the cytokines IL-17, IL-1, and TNF, especially when the infection was caused by *L. (V.) braziliensis* [22,23]. The main sources of IL-17 are Th17-RORγT+ cells, and their role in the immune response against *Leishmania* spp. has already been reported by other authors [22,23,24]. These cells may have ambiguous properties depending on which species of Leishmania are causing the infection, as in VL they are involved with parasite clearance and in CL are related to the exacerbation of the lesions and disease progression [24].

The generation and maintenance of memory T cells are the most important aspects of a vaccine [25]. The resolution of Leishmania infection can generate long-term immunity mediated primarily by a short-lived population of specific CD4+ T cells, which are maintained by a small number of parasites that persist after the patient is cured [26]. Memory T cells are generated with different functions and phenotypes. Central memory T cells (TCM—CD45RA-CD62L+) have been described as a memory cell subpopulation that can recirculate through secondary lymphoid organs and function as a reservoir of antigen-specific T cells. These cells proliferate and differentiate, migrate to the site of injury, and protect the individual against infection [27]. The effector memory T cells (TEM – CD45RA+CD62L-) are cells that recirculate between the blood and the body and can respond rapidly against reinfection by producing effector cytokines [28]. However, the frequency of memory T cells generated during or after CL in humans and their cognate epitopes is still not fully known.

Through reverse vaccinology [29], we have recently identified in the predicted proteome of *Leishmania (Viannia) braziliensis* CD4+ and CD8+ T cell epitopes [30,31]. In the present study, 2 groups of 5 peptides each were used to evaluate their potential to restimulate in vitro memory subsets of CD4+ and CD8+ of CL and healthy control individuals. Additionally, the proliferation capacity of memory subsets, T-bet and RORγT expression in CD4+ and CD8+ T cells were measured, and cytokines produced and released by stimulated cells. Our findings indicate that during CL there is an increase in the frequency of TEM CD8+ cells and of TEM and TCM CD8+ cells post-treatment. Using the peptide pools, we did not observe differences in the frequency of memory cells and/or cell proliferation during CL. We observed more cell proliferation post-treatment suggesting the presence of memory T cells reactivated upon peptide stimulation. Our cytokine data reinforces the Th1-based profile investigated and we conclude that the epitopes used in this study are not cognates of memory T cells. However, effector memory T cells were found during disease and central memory after disease resolution.

## 2. Materials and Methods

### 2.1. Ethical Aspects

This work was evaluated and approved by the Ethics Committee in Human Research from the Aggeu Magalhães Institute (Approval code: 11083812.7.0000.5190). All the individuals agreed to participate in the study and signed the Term of Free and Informed Consent.

### 2.2. Patients and Samples

CL patients were divided into active disease (AD = 7), meaning those who did not receive treatment at the time of sample collection, and post-treatment (PT = 8), meaning those who had received it. The third and last group consisted of healthy individuals (CT = 15). Each experiment was performed in triplicates. The patients’ inclusion criteria were based on the standards recognized by the Reference Service of the Aggeu Magalhães Institute (Oswaldo Cruz Foundation—FIOCRUZ) and the Dermatology Clinic of the Federal University of Pernambuco, which included the presence of active lesions, absence of treatment for Leishmaniasis, and the positive diagnosis in two different tests (i.e., molecular and parasitological). Exclusion criteria were the absence of lesions, the presence of other dermatological conditions, and previous treatment for Leishmaniasis.

### 2.3. Blood Harvesting and PBMC Isolation

Approximately 40 mL of peripheral blood from individuals of all groups (AD, PT, and CT) were collected in heparinized tubes, diluted in a proportion of 2:1 (*v*/*v*) with phosphate-buffered saline (PBS) (pH 7.2), and submitted to a Ficoll gradient solution (Amersham Biosciences, Uppsala, Sweden). After centrifugation at 400× g for 30 min at 20 °C, the buffy coat containing the peripheral blood mononuclear cells (PBMCs) was obtained. Cells were washed with PBS and submitted to centrifugation twice (300× *g* for 10 min at 20 °C). PBMCs were resuspended with 2 mL of RPMI 1640 medium (Cultilab, Campinas, SP, Brazil) supplemented with 10% fetal bovine serum (FBS) (Cultilab, Campinas, SP, Brazil) and 1% of antibiotics (100 UI/mL penicillin and 100 μg/mL streptomycin- Sigma, St. Louis, MO, USA). Cell viability was assessed using the Trypan blue dye (Sigma, St. Louis, MO, USA) and counted using a hemocytometer. The final concentration was adjusted to 106 cells for subsequent staining.

### 2.4. Parasite Growth

*Leishmania (Viannia) braziliensis* (MHOM/BR/75/M2903) were cultivated in T-25 cell culture flasks containing complete (10% FBS, 5% penicillin-streptomycin, 5% glutamine) Schneider’s Drosophila medium (Sigma-Merck, Darmstadt, Germany) at 27 °C in an incubator under sterile conditions. Parasite infectiveness was maintained through animal passage and a maximum of five culture passages were performed prior to expanding the culture to obtain the total soluble antigen.

### 2.5. Peptide Preparation

The ten peptides used in this work were restricted to human MHC supertypes and were mapped to four different proteins from *Leishmania braziliensis* which are conserved among Leishmania species [30]. Eight peptides were derived from three hypothetical proteins: four from LbrM.34.3630 (peptides 1, 4, 5, and 8); three from LbrM.01.0110 (peptides 6, 7, and 10), found in both Trypanosoma and Leishmania species but with yet undefined properties or other features; and one from LbrM.06.0820 (peptide 3), a hypothetical protein found only in Leishmania and closely related groups but, so far, absent in Trypanosoma species. The last two peptides (2 and 9) were derived from LbrM.14.1680, a protein annotated as a putative Synaptojanin, an inositol/phosphatidylinositol phosphatase involved with vesicle-mediated transport. Peptides derived from the same protein share substantial overlaps in sequence, differing by one or two amino acid residues (peptides 1 and 4, for example, have 14 residues in common, as do peptides 4 and 5, all from LbrM.34.3630). These peptides were tested for their immunogenicity in a previous work by our group (31) and it was possible to observe a trend when these peptides stimulated healthy individual cells vs. patient cells. Two pools of peptides were thus created: P1, which consisted of peptides 2, 4, 8, 9, and 10 or the “unspecific pool”, was able to indistinctly stimulate cells from healthy individuals and CL patients; and P2, comprised of the peptides 1, 3, 5, 6, and 7, was able of specifically stimulate samples from CL patients, and was, therefore, considered as the “specific pool” (Table 1).

### 2.6. PBMC In Vitro Stimulation with Peptide Pools

This assay was performed based on a previous work by our group (31). The peptides corresponding to the unspecific or specific groups (P1 or P2) were associated in one single tube in a stock concentration of 1 mg/mL. 2 × 10^5^ PBMCs from all evaluated groups (AD, PT, and CT) were deposited per well in 96-U-bottom well plates (BD Falcon, USA) and stimulated individually with 15 μg/mL of the peptide pools P1 or P2. Each pool was tested in triplicates per group. The plates were incubated at 37 °C with 5% CO_2_ for 96 h. After that period, the plates were centrifuged for 10 min at 400× *g*, the supernatants were collected and stored at −80 °C for cytokine evaluation and cells were retrieved for the subsequent tests.

### 2.7. Antibodies and Flow Cytometry of CD4+ and CD8+ T Cells

A panel of memory T cells was used with the following antibodies purchased from BD Biosciences: CD3 (clone: SK7), CD4 (clone: SK3), CD8 (clone: RPA-T8), CD62L (clone: TWAJ), and CD45RA (clone: HI100). Central memory T cells were characterized by the following markers: CD4+/CD8+ CD45RA-CD62L+ and effector memory T cells: CD4+/CD8+ CD45RA+CD62L-. Besides that, the expression of the transcription factors RORγT (clone: Q21-559) and T-bet (clone: O4-46) was also assessed. For that, the PBMCs were washed with PBS-W (PBS with 0.5% FBS (Gibco) and 0.1% sodium azide (Sigma, St. Louis, MO), centrifuged (400× *g*, for 10 min at 20 °C), and transferred to polystyrene tubes. The extracellular antibodies were then added: anti-CD3APCH7, anti-CD4 FITC, anti-CD8 APC, anti-CD45RA PerCP-Cy5.5, and anti-CD62L PE, and the cells were incubated for 30 minutes at 20 °C. After that period, the cells were washed with PBS-W, fixed, and permeabilized using the Cytofix Cytoperm solution (BD Biosciences, San Jose, CA, USA) for 40 min at 20 °C. Then, cells were washed with the Perm-Wash buffer (BD Biosciences, San Jose, CA, USA) (400× g for 5 min) and incubated with the intracellular antibodies anti-T-bet PE CF594 and anti-RORγT BV421 for 30 min at 20 °C. After incubation, cells were washed one more time, resuspended in a fixing solution containing 1% of paraformaldehyde, and acquired by flow cytometry. Gating strategies for all flow cytometry assays are provided in the Appendix A.

### 2.8. CFSE Cell Proliferation Assay

One million PBMCs per patient were stained with Carboxyfluorescein succinimidyl ester (CFDA-SE, Invitrogen, Waltham, MA, USA) to evaluate the cell proliferation levels induced by the peptide pools. The cells were resuspended in 1 mL of PBS (pH 7.2) containing CFDA-SE and incubated at 37 °C for 10 min. After incubation, the reaction was quenched with 1 mL of ice-cold RPMI 1640 (4 °C) containing 2 mM L-glutamine, 50 mg/L of gentamicin sulfate, and supplemented with 10% fetal calf serum (FCS) (Gibco, New York, NY, USA). Then, the cells were centrifuged and washed with PBS, being resuspended in 1 mL supplemented RPMI 1640 at a density of 2 × 10^5^ cells/mL. The cells were plated with 15 μg of each peptide pool. The plates remained for 96 h in a 5% CO_2_ incubator under 37 °C. This timepoint was previously standardized by our group (30, 31) after performing different kinetics assays. After this period, the cells were harvested, deposited in polystyrene tubes, washed with PBS (pH 7.2), and analyzed by flow cytometry. For gate optimization, baseline cells (unstimulated) were used as the negative control, and as the positive control, the mitogen phytohemagglutinin (PHA).

### 2.9. Cytokine Measurement after Peptide Pools Stimulation

For this assay, cell culture supernatants were harvested at 48 h and were used to quantify cytokines of the Th1 (IL-2 and IFN-γ) and Th2 profiles (IL-4 and IL-10) using the ELISA kit Human Th1/Th2 Uncoated ELISA Kit (Invitrogen, Thermo Fischer Scientific) following the manufacturer’s instructions.

### 2.10. Flow Cytometry Acquisition and Statistical Analyses

Flow cytometry experiments were performed on a FACSCalibur using CELLQuestProTM for the CFDA-SE assay and on a FACSAria III using FACSDiva (both from BD Biosciences, San Jose, CA, USA) for the immunophenotyping. Fluorescence of 20,000 lymphocyte-gated events was acquired based on scatter parameters of size and granularity. The data was analyzed and treated with FlowJo v10.1 (Tree Star Inc., Ashland, OR, USA). For CFDA-SE, unstimulated cells were used during the analysis to set quadrant parameters and the basal level of lymphocyte proliferation, as a positive control of proliferation, PHA was used. For the T-bet and RORγT assays, labeled cells from control individuals were used to set the quadrant standards. Fluorescence minus one (FMO) control was used to avoid background signals and set the gates accurately. The statistical analysis was performed using GraphPad Prism Software v.7 and the data was analyzed with the non-parametric Mann–Whitney U-test. The differences were considered statistically significant when *p* ≤ 0.05.

## 3. Results

### 3.1. T-Bet and RORγT Expression by CD4 and CD8 T Cells of Active Disease and Post-Treatment Patients after Peptide Pools Stimulation

The master transcription factors T-bet (Th1 cells) and RORγT (Th17 cells) were measured in CD4+ and CD8+ T cells from the evaluated groups. As expected, the control group did not present high levels of these markers since there was no disease background to support that. On the other hand, a significant difference was observed regarding the production of T-bet by both subsets (CD4+ and CD8+T cells) from the CL group after stimuli with the peptide pool P1 and in the presence of the soluble Leishmania antigen (SLA). Besides, it was also possible to detect a higher expression of these markers when the same cells were stimulated with the specific pool P2. This could be explained by the active disease state of the patients, which leads to the expansion and differentiation of naïve into effector cells that have pro-inflammatory features. Regarding the PT group, the CD4+ T cell population had a significant increase in RORγT expression compared to the control group after stimulation with P2, and an augment in T-bet expression was observed in this group for CD4+ and CD8+T cells. This fact might be attributed to immunological memory since the cells were exposed to an antigen that the patients’ immune systems have seen before (Figure 1). 

### 3.2. Central and Effector Memory T Cell Reactivation with Peptide Pools

To identify the two main subtypes of memory T cells (central memory—TCM and effector memory—TEM) after stimuli, the following panel was used: TCM—CD4+/CD8+CD45RA-CD62L-, and TEM—CD4+/CD8+CD45RA+CD62L-. The frequency of CD4+ TCM cells was comparable among all the evaluated groups, with or without stimuli. CD62L is a marker that is also present in naïve populations, which might explain the fact that the control group had an increased number of cells with this phenotype. However, it is important to mention that the PT group had the highest frequency of TCM cells compared to the CL group, supporting the fact that these cells have memory because they were previously exposed to Leishmania antigens. On the other hand, CD8+TCM cells from both CL and PT groups had significantly higher frequencies with and without stimuli when compared to the CT group, which might be caused by these cells having a high proliferative capacity after antigen stimulation, especially if they have been exposed to the same antigen source before. Regarding CD4+TEM cells, there was no significant difference among the evaluated groups. However, for CD8+TEM cells, there was a significant difference after stimuli with the peptide pool P1 on the CL group compared to the CT and a higher frequency of these cells was overall observed in the PT group when compared to the CT (Figure 2). 

### 3.3. PBMCs Proliferation Profile after Peptide Pool Stimulation

After the 96 h staining with CFSE, the percentage of proliferated cells was evaluated by flow cytometry. Cells from the CL and CT groups without stimulus (Unstim) and with 15 μg of each peptide pool (P1 or P2) or 15 μg of the soluble Leishmania antigen (SLA) were assessed. As the positive control of proliferation, 20 μg/mL of the mitogen phytohemagglutinin (PHA) was used (data not shown). As previously mentioned, P1 has an unspecific characteristic since it is able to induce the proliferation of both healthy and infected individuals. This fact was corroborated in the present study, as it is possible to observe an increase in the cell frequency in all evaluated groups after this stimulus. Overall, both patient groups CL and PT had increased cell proliferation compared to the CT, which was expected because of the acute state of the disease in the CL group and the re-exposure to the antigen in the PT group, which leads to cell expansion and activation (Figure 3). 

### 3.4. Cytokine Production of CL and CT PBMCs after Peptide Pool Stimulation

For this assay, cytokines of the Th1 (IL-2 and IFN-γ) and Th2 profiles (IL-4 and IL-10) were measured. At 24 and 72 h, only the cytokine IL-10 was detected in the CL group (data not shown), however, no significant differences were observed after comparing stimulated vs. unstimulated cells. On the other hand, at 48 h, the cytokines IL2, IFN- γ, IL-4, and IL-10 were detected with both peptide pools and the SLA stimuli on patients before treatment. There was a trend of higher production of these cytokines after stimulation with the peptide pools when compared to the unstimulated cells or those stimulated with SLA. These levels were more pronounced for IL-10 and IL-4, though 1 patient presented a high level of IL-2 after P1 stimulation, and 3 patients presented high levels of IFN-γ after P2 stimulation (Appendix A). Regarding the CT group, at the same time point of 48 h, the cytokines IL-2, IL-10, and IFN-γ were detected. Baseline cells (unstimulated) presented high levels of these molecules, and the stimuli that they received did not have a meaningful impact on the overall cytokine production, which was already expected (Appendix A).

## 4. Discussion

A safe and effective vaccine for Leishmaniasis has been extensively investigated over the years and, currently, there are no globally licensed vaccines for human CL. Some candidates have been advanced for clinical trials but had inconsistent results between placebo and control groups [32,33]. With the perspective to evaluate the capacity of anti-Leishmania epitopes to stimulate memory T cells and of identifying potential new candidates for CL, the present work was idealized. Ten peptides of *Leishmania (Viannia) braziliensis*, which were previously identified using in silico tools [30,31], were the candidates of choice. These peptides were classified into two groups: P1, an unspecific pool, stimulates indistinctly cells from healthy individuals and CL patients; and P2, a specific pool, stimulates specifically only CL patients’ cells.

The efficacy of licensed vaccines against Leishmaniasis remains controversial. Two were approved to be used in humans, one of which uses killed *Leishmania amazonensis* parasites as an immunotherapeutic approach in Brazil [34], and the other utilizes the inoculation of live *Leishmania major* parasites to induce a durable immunological response, used in Uzbekistan [35]. The other licensed vaccines were developed for the prophylactic immunization of dogs with recombinant proteins [36,37,38]. While immunotherapeutic approaches need to be combined with the conventional treatment [39], those which use live parasites must be used with extreme caution, since they might produce chronic and resistant lesions. This fact brings up concerns, especially regarding risk groups such as immunocompromised patients, the elderly, pregnant women, and children [11]. 

It is well described in the literature that the transcription factor T-bet is the main regulator of the Th1 profile, considered the protective response against intracellular pathogens [40]. However, when it comes to Leishmania infections, its role is still not completely understood. The present work suggests that both peptide pools were able to induce a significantly higher expression of T-bet in CL patients before and after treatment. Besides, the peptides which compose the pools were selected using tools that can predict their ability to stimulate CD4+ and CD8+T cells [30], suggesting that they were recognized and redirected these cell populations towards a Th1 profile. 

The transcription factor RORγT is associated with the differentiation of T helper cells to the Th17 profile [41]. It has already been demonstrated that CD8+T cells express RORγT in inflammatory diseases [42,43]. Recent studies in the context of CL corroborate this information [44,45,46,47], however, the role of the peripheral response remains poorly understood. An important addition is that this phenotype also seems to be dependent on the Leishmania species [48], which makes this characterization in PBMCs an important avenue of investigation. Although not statistically significant, the results point to a trend of higher RORγT expression in CD8+T cells after stimulation with the P1 pool. This can be possibly explained because this pool induces an exacerbated proliferation of cells from both healthy individuals and CL patients, which might explain this potent pro-inflammatory profile.

For a vaccine to be considered efficient, it is crucial that it induces immunological memory [49,50]. Memory T cells with different functions and phenotypes are generated during the immune response and two main populations are commonly identified: TCM, which act as a reservoir of antigen-specific cells, and TEM which are capable of rapidly migrating to the affected tissue and producing cytokines [51]. When these phenotypes were evaluated, it was possible to observe a trend towards a higher frequency of CD4+ TEM in patient cells stimulated with either the peptides or the SLA. Theoretically, the CL group did not have prior contact with the parasite although they reside in endemic regions, this would explain the low frequency of effector memory cells. For CD8+ TCM, on the other hand, a significant difference was observed between CL vs. CT and PT vs. CT after stimulating their cells with P1, P2, and the positive control SLA. It is well described that TCM cells have a higher proliferative capacity in comparison to TEM because they are centralized in secondary lymphoid organs [52]. Therefore, the results from this study are in line with this evidence since the TCM proportion was higher than TEM. Another explanation for our finding is that the cognate cells for our epitopes are effector cells rather than memory cells, or even that the amount of memory cells is low and could not be detected through in vitro re-stimulation in these time points.

A previous work by our group [31] evaluated the immunogenicity of each peptide individually. It was observed then that there were no surprising differences between patients before treatment and healthy individuals. This was one of the reasons why in the present work we have decided to use them pooled together. One other motivation was that it is well reported that T cells respond more efficiently and in a long-lasting manner to antigens that are presented in sufficient amounts for enough time [53]. Other groups also reported a high cell proliferation rate after stimulation with the soluble Leishmania antigen in patients with active disease [54,55]. 

With the results obtained in the present work, it was not possible to observe a significant difference in the percentage of cell proliferation from patients in comparison with the CT group. However, there was a trend of the stimulated cells having higher proliferation rates compared to the unstimulated ones. The CT group, when stimulated with P1, induced a higher proliferation than the unstimulated condition, and that might be because this pool stimulates cells in a generalized way. In the CL group, there was no significant difference, and this could be due to the active disease state of the patients, which leads to proliferation regardless of the other stimuli. 

In the classical form of CL, cytokines such as TNF and IFN-γ seem to play roles that lean in between pro- and anti-inflammatory, which highlights the complexity of this response [56,57,58]. Pro-inflammatory cytokines from the Th1 profiles, such as IFN- γ, are associated with macrophage activation and nitric oxide production that promotes parasite elimination. On the other hand, its exacerbation might lead to the enhanced pathogenesis of the disease, causing chronic lesions [59]. In humans, Th2 cytokines such as IL-10 are related to immunosuppression and pathology in CL [60,61], but they are required to counterbalance widespread inflammation. In this sense, we have proposed a survey evaluation of these responses by quantifying these cytokines on the cell supernatants from patient samples before treatment.

There were no significant differences regarding cytokine production in the CL vs. CT groups (*p*-value > 0.05), however, increased levels of IL-10 and IL-4 were detected in the CL group after the cells were stimulated with both peptide pools. Since ROR-γT was particularly augmented in this set of patient cells, this could be a regulatory/compensatory measure of the immune response to control a possible exacerbated inflammation. IFN-γ and IL-2 were detected in a small number of patient cells, which might be particularly related to the number and size of their lesions, explaining the Th1-skewed response. Regarding the CT group, unstimulated and stimulated cells had comparable levels of cytokines, which was already expected for they were not exposed to the pathogen before and there was no immune response mounted against the antigens. Additionally, this pattern of response might have been observed due to the intrinsic characteristics of each individual immune system. Therefore, we hypothesize that we might have worked with a small heterogeneous group, which delivered distinct results among themselves.

A limitation of this study is that not all T cell memory markers described so far were included in our flow cytometry panel, and intracellular cytokine staining was not performed to corroborate with the ELISA results. 

Promising CD4+ and CD8+ epitopes were evidenced herein, and it opens perspectives for using the peptide pools as an immune active formulation or a vaccine. Additionally, it is worth emphasizing that no adjuvant molecule has been used in the experiments performed and the combined use of peptides and adjuvants could provide even more robust responses. A more thorough evaluation with a higher number of samples to identify different cell profiles, cytokine production, and an in vivo assay needs to be performed to attest that these peptides would be good vaccine candidates. However, the results presented here have the potential to point toward the development of a safe and effective vaccine against CL.

## Figures and Tables

**Figure 1 vaccines-11-00158-f001:**
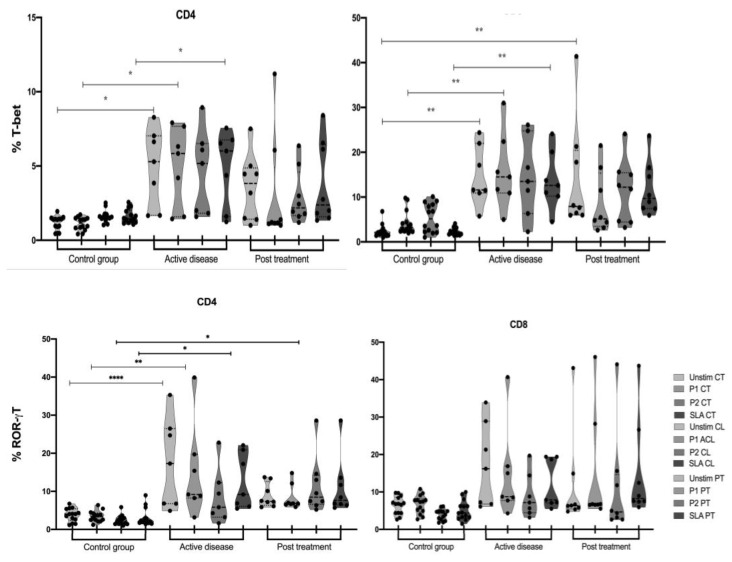
T-bet and RORγT expression levels in patient and control cells. CT—control group, CL—active disease group, PT—post-treatment group, Unstim—unstimulated cells, P1—pool 1, P2—pool 2, SLA—Soluble Leishmania antigen. *p*-values: * ≤ 0.01; ** ≤ 0.003; **** < 0.0001.

**Figure 2 vaccines-11-00158-f002:**
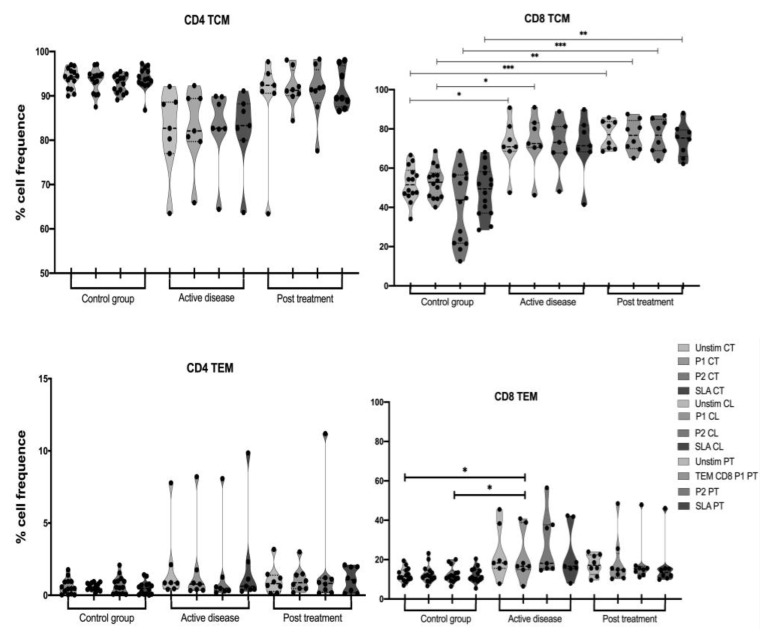
Memory profile of patients’ and controls’ CD4 and CD8 T cells after peptide stimulation. CT—control group, CL—active disease group, PT—post-treatment group, Unstim—unstimulated cells, P1—pool 1, P2—pool 2, SLA—Soluble Leishmania antigen. *p*-values: * ≤ 0.05; ** ≤ 0.008; *** ≤ 0.0005.

**Figure 3 vaccines-11-00158-f003:**
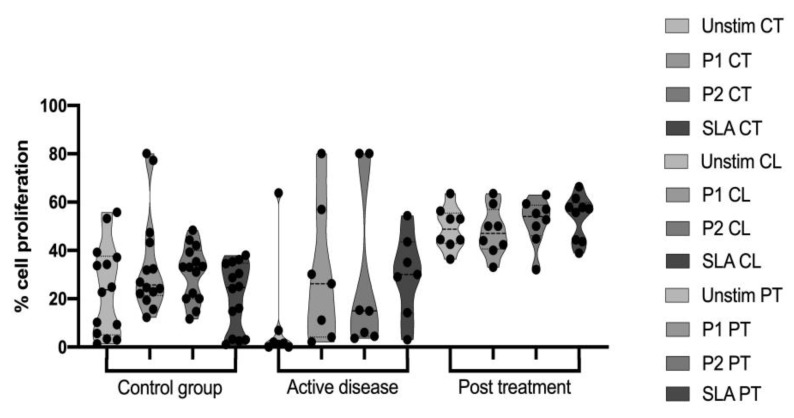
CFSE cell proliferation assay after peptide stimulation. CT—control group, CL—active disease group, PT—post-treatment group, Unstim—unstimulated cells, P1—pool 1, P2—pool 2, SLA—Soluble Leishmania antigen. *p*-value > 0.05.

**Table 1 vaccines-11-00158-t001:** Peptide sequences and protein identifiers of each pool.

Pool 1	Peptide Sequence	Protein ID
P2	GNDHYYEYILWKYHG	LbrM.14.1680
P4	PFLYYYILCYARDFG	LbrM.34.3630
P8	HNAPFLYYYILCYAR	LbrM.34.3630
P9	NDHYYEYILWKYHGA	LbrM.14.1680
P10	TVFYTISFDQMERYL	LbrM.01.0110
**Pool 2**	**Peptide sequence**	**Protein ID**
P1	FLYYYILCYARDFGS	LbrM.34.3630
P3	ISFEIYPAHLFYSLI	LbrM.06.0820
P5	APFLYYYILCYARDF	LbrM.34.3630
P6	VFYTISFDQMERYLA	LbrM.01.0110
P7	YTISFDQMERYLAAI	LbrM.01.0110

## Data Availability

Not applicable.

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
