# Peer review of "Central and Effector Memory Human CD4+ and CD8+ T Cells during Cutaneous Leishmaniasis and after In Vitro Stimulation with Leishmania (Viannia) braziliensis Epitopes"

_vaccines, 2023, doi:10.3390/vaccines11010158_

Round 1

Reviewer 1 Report

The study is an extension of some previous work of the authors on cutaneous leishmaniases (CL) where they identified T cell epitopes of Leishmania (Viannia) braziliensis which stimulated the patients T cells in vitro. This study reports a potentially immunogenic profile of the peptide pools which can be used in the development of formulations against Leishmania. Briefly, the changes in the master transcription factors T-bet and RORγT expression levels and cytokines have been depicted in patient and control cells. The memory profile (patients’ vs controls) of CD4 and CD8 T cells after peptide stimulation and cell proliferation are shown in Fig. 2 and 3, resp. At the outset, the study design is fine, results are convincing.

Specific remarks:

(1) Was there any exclusion/inclusion criteria other than recruiting patients infected with CL? What was the criteria for sample size?

(2) Th1 (IL-2 and IFN-γ) and Th2 cytokines (IL-4 and IL-10) were measured by ELISA. What was the accuracy/variations in the method when run in triplicate for the same sample?

(3) TCM serves as a reservoir of antigen-specific T cells. TEM, the effector memory T cells, on the other hand, help respond the body rapidly against reinfection by producing effector cytokines. Authors report an increase in TCM and TEM frequency in CL and suggest that selected peptide pools were able to induce the expression of T-bet in CL patients before and after treatment, besides an increase in ‘cell proliferation post treatment suggesting the presence of memory T cells reactivated upon peptide stimulation’. However, the manuscript lacks a mechanistic approach explaining the said changes in response to the peptides at the structural level. This may add value to the study.

(4) Any toxicity study on the peptides? In vitro, or in animal model?

(5) Authors may like to add a small paragraph on the limitations of the study.

Other minor issues:

(i)                 The terms TCM and TEM should be defined in the abstract. Give full forms.

(ii)               The abbreviation CL should be defined in the manuscript section/ introduction where it is first used. It will be good to start the introduction (first line) with an introduction to CL, instead of abruptly starting with ‘Leishmania parasites are transmitted to humans through blood meals of infected sandflies’. Authors may like to update Ref. 2 (WHO Leishmaniasis fact sheet).

(iii)             Please use the correct symbol to indicate â—¦C (Line 129).

(iv)             Always italicize the spp. name (Line 135, 138, and elsewhere).

(v)               Are the structures of the peptides which were isolated (Section 2.5) available?

(vi)             Line 170. What is 2 x 105? May correct.

(vii)           Line 173, please correct CO2.

(viii)         Language: Language is fine, but authors may read the MS carefully for errors, such as Line 96 (did not observed’) and similar typographical/ grammatical mistakes.

Author Response

Point 1: Was there any exclusion/inclusion criteria other than recruiting patients infected with CL? What was the criteria for sample size?

Response 1: We thank the reviewer for the constructive criticism. Apart from the presence of active lesions and no previous treatment for Leishmaniasis, the inclusion criteria for CL patients were based on the positive diagnosis in two different tests (molecular and parasitological). This was in line with the standards preconized by the Reference Service of the Aggeu Magalhães Institute (Oswaldo Cruz Foundation) and the Dermatology Clinic of the Federal University of Pernambuco. Since most of the patients live in rural areas which are difficult to access, having a robust sample size is, in most cases, challenging. Another reason why it is hard to acquire samples is that patients do not seek for medical assistance immediately, so the frequency of cases per month is low. However, our aim was including at least 10 total patient samples which would be paired with the healthy controls, what was successfully done (15 total patient samples + 15 controls). We believe that with these numbers we achieved a solid statistical analysis in our work. The exclusion criteria were the absence of lesions, presence of other dermatological conditions and previous treatment for Leishmaniasis. All these criteria were added to the methodology section (lines 114-120).

Point 2: Th1 (IL-2 and IFN-γ) and Th2 cytokines (IL-4 and IL-10) were measured by ELISA. What was the accuracy/variations in the method when run in triplicate for the same sample?

Response 2: We thank the reviewer for the question but there were no major variations within the triplicates of the detected cytokines. The coefficient of variation was within the range of 20%.

Point 3:  TCM serves as a reservoir of antigen-specific T cells. TEM, the effector memory T cells, on the other hand, help respond the body rapidly against reinfection by producing effector cytokines. Authors report an increase in TCM and TEM frequency in CL and suggest that selected peptide pools were able to induce the expression of T-bet in CL patients before and after treatment, besides an increase in ‘cell proliferation post treatment suggesting the presence of memory T cells reactivated upon peptide stimulation’. However, the manuscript lacks a mechanistic approach explaining the said changes in response to the peptides at the structural level. This may add value to the study.

Response 3: We agree with the reviewer’s comment and our data suggests that, compared to the control samples, the post-treatment stimulated cells are significantly increased on TCM, and higher but not statistically significant on the TEM compartment. The cell proliferation data suggests that these subsets are possibly reactivated cells, however, as we discuss in lines 381-384, “another explanation for our finding is that the cognate cells for our epitopes are effector cells rather than memory cells, or even that the amount of memory cells is low and could not be detected through in vitro re-stimulation in these time points”. The higher expression of T-bet after treatment suggests that these cells have effector phenotype, and further investigations with a bigger sample size including more memory markers and in vivo assays would be helpful to validate this hypothesis.

Point 4: Any toxicity study on the peptides? In vitro, or in animal model?

Response 4: We believe this is a very important comment. However, no toxicity studies were performed in the present work, but according to the viability of the cells in culture, and in a pilot animal study (data unpublished), the pooled peptides do not present toxicity against host cells, but rather stimulate a pro-inflammatory response.

Point 5: Authors may like to add a small paragraph on the limitations of the study.

Response 5: We thank the reviewer for the suggestion and the main limitations identified were added in this new version of the manuscript (lines 424-426).

Other minor issues:

Point I:                 The terms TCM and TEM should be defined in the abstract. Give full forms.

Response I: We thank the reviewer for the observation. The acronyms were broken down on the text.

Point II:              The abbreviation CL should be defined in the manuscript section/ introduction where it is first used. It will be good to start the introduction (first line) with an introduction to CL, instead of abruptly starting with ‘Leishmania parasites are transmitted to humans through blood meals of infected sandflies’. Authors may like to update Ref. 2 (WHO Leishmaniasis fact sheet).

Response II: We thank the reviewer for the suggestions. The abbreviation was written properly, and the introduction and reference were adjusted.

Point III:             Please use the correct symbol to indicate â—¦C (Line 129).

Response III: We thank the reviewer for the observation. The symbol was fixed.

Point IV:             Always italicize the spp. name (Line 135, 138, and elsewhere).

Response IV: We thank the reviewer for the observation. The full species names were italicized, however, the “spp.” abbreviations were not, since it is just an abbreviation of “species” instead of a taxonomic rank.

Point V:               Are the structures of the peptides which were isolated (Section 2.5) available?

Response V: The bioinformatics and docking analyses of the individual peptides can be found in the previous work of our group (Freitas-e-Silva et al, 2016). The work was referenced on line 144.

Point VI:             Line 170. What is 2 x 105? May correct.

Response VI: The scientific notation was corrected.

Point VII:           Line 173, please correct CO2.

Response VII: This adjustment was made.

Point VIII:         Language: Language is fine, but authors may read the MS carefully for errors, such as Line 96 (did not observed’) and similar typographical/ grammatical mistakes.

Response VIII: We thank the reviewer for the observation. The manuscript was reviewed, and typos and grammatical mistakes were corrected.

Reviewer 2 Report

In this manuscript investigators have screened pool of peptides from Leishmania braziliensis in vitro that can potentially be used as vaccine candidates. The study is well conducted and I have no further comments on this manuscript.

Author Response

We thank the reviewer for the comments.

Reviewer 3 Report

The manuscript presented by Beatriz Coutinho de Oliveira et al sheds light on a potentially immunogenic profile of the peptide pools, which can support the development of anti-Leishmania formulations. Most experiments are well controlled. Some conclusions are not support by evidence. Below are some comments to help strengthen the manuscript.

1.     Legends are way too vague, don’t emphasize the quality of their work and make difficult to follow the flow and rational of the work. It is better to show the statistical analysis of this paper in figure legends.

2.     This dataset suggests that during CL there is an increase in the frequency of TCM and TEM, especially on the CD8 compartment. It is better to show the IL2, IFN γ, IL-4, IL10 expression in CD8+ and CD4+ with or without stimulation by Flow cytometry.

3.     For supp.3 and supp 4, no statistical analysis.

Author Response

Point 1: Legends are way too vague, don’t emphasize the quality of their work and make difficult to follow the flow and rational of the work. It is better to show the statistical analysis of this paper in figure legends.

Response 1: We thank the constructive comment of the reviewer. In this new version of the manuscript, we added the p-values to the figure legends to make it clear to readers.

Point 2: This dataset suggests that during CL there is an increase in the frequency of TCM and TEM, especially on the CD8 compartment. It is better to show the IL2, IFN γ, IL-4, IL10 expression in CD8+ and CD4+ with or without stimulation by Flow cytometry.

Response 2: We thank the reviewer for this interesting suggestion. We have also imagined that this step could be important to address. However, due to limited resources and funding for this specific project, it was not possible to perform the cytokine evaluation by flow cytometry. In order to address this point critically in our work, we have added to the limitation paragraph on lines 424-426.

Point 3: For supp.3 and supp 4, no statistical analysis.

Response 3: We thank the reviewer for this observation, and we emphasize that the statistical analysis (one-way ANOVA) was performed; however, the p-value was >0.05 for all comparisons. Now we added the statistical analyses to the line 412.

Round 2

Reviewer 1 Report

I accept the responses and believe all the responses gave been incorporated in the revised version. Authors may certify.